# Pediatric Fatty Liver and Obesity: Not Always Just a Matter of Non-Alcoholic Fatty Liver Disease

**DOI:** 10.3390/children5120169

**Published:** 2018-12-13

**Authors:** Renata Alfani, Edoardo Vassallo, Anna Giulia De Anseris, Lucia Nazzaro, Ida D’Acunzo, Carolina Porfito, Claudia Mandato, Pietro Vajro

**Affiliations:** 1Pediatrics Residency Joint Programs, University of Naples Federico II, 80131 Naples, Italy; renata.alfani@gmail.com (R.A.); edoardo.vassa@gmail.com (E.V.); ida.dacunzo@gmail.com (I.D.); carolinaporfito@gmail.com (C.P.); 2Pediatrics Residency Joint Programs, University of Salerno, 84081 Baronissi (Salerno), Italy; 3Clinical Pediatrics Azienda Ospedaliera Universitaria San Giovanni di Dio e Ruggi D’Aragona, 84131 Salerno, Italy; annagiulia.deanseris@gmail.com (A.G.D.A.); nazzaroluci@libero.it (L.N.); 4Children’s Hospital Santobono-Pausilipon, Department of Pediatrics, 80129 Naples, Italy; cla.mandato@gmail.com; 5Department of Medicine, Surgery and Dentistry, Scuola Medica Salernitana, Pediatrics Section, University of Salerno, 84081 Baronissi (Salerno), Italy

**Keywords:** fatty liver, obesity, NAFLD, differential diagnosis, systemic disorders, genetic and metabolic disorders

## Abstract

Obesity-related non-alcoholic fatty liver disease (NAFLD) represents the most common cause of pediatric liver disease due to overweight/obesity large-scale epidemics. In clinical practice, diagnosis is usually based on clinical features, blood tests, and liver imaging. Here, we underline the need to make a correct differential diagnosis for a number of genetic, metabolic, gastrointestinal, nutritional, endocrine, muscular, and systemic disorders, and for iatrogenic/viral/autoimmune hepatitis as well. This is all the more important for patients who are not in the NAFLD classical age range and for those for whom a satisfactory response of liver test abnormalities to weight loss after dietary counseling and physical activity measures cannot be obtained or verified due to poor compliance. A correct diagnosis may be life-saving, as some of these conditions which appear similar to NAFLD have a specific therapy. In this study, the characteristics of the main conditions which require consideration are summarized, and a practical diagnostic algorithm is discussed.

## 1. Introduction

Obesity-related non-alcoholic fatty liver disease (NAFLD) has become one of the most common causes of pediatric liver disease, as a result of the parallel epidemics of overweight and childhood obesity [1]. An increase in childhood obesity has led to a rise in prescriptions of liver ultrasounds (US) and liver function tests, which, when abnormal, frequently result in a consequential presumptive diagnosis of NAFLD. However, it is necessary to maintain a high index of suspicion and make an accurate differential diagnosis to rule out several other obesity-unrelated or coexistent liver diseases. This is all the more important for those patients who are not in the NAFLD classical age range and for those for whom a satisfactory response of liver test abnormalities to weight loss after dietary and physical activity counseling cannot be obtained or even verified due to poor compliance. There are in fact a sizeable number of diseases involving fatty liver/hypertransaminasemia, which are rare when considered individually, but represent a large group if considered collectively and can mimic NAFLD in an obese child. Since some of them may sometimes require life-saving specific therapies, here we summarize the characteristics of the main conditions which require consideration no the basis of a practical diagnostic algorithm.

## 2. NAFLD

The term NAFLD refers to a wide spectrum of histological hepatic lesions ranging from simple (usually macrovesicular) steatosis to non-alcoholic steatohepatitis (NASH), cirrhosis, and hepatocellular carcinoma as frequently seen in obese individuals (obesity-related liver disease). Its histologically proven prevalence in children in the United States (as revealed in autopsies after accidents) ranges from 9.6% in normal-weight individuals to 38% in obese ones. Due to its tendency to progress through this range in childhood or after the transition to adulthood, early diagnosis and treatment are important issues at all ages. Treatment should address not only the liver disease itself but also the entire spectrum of comorbidities to improve overall survival and quality of life [2]. The European Society of Pediatric Gastroenterology Hepatology and Nutrition (ESPGHAN) expert committee recommends that differential diagnoses of NAFLD should primarily be based on clinical features and ultrasonographic/blood tests. In ambiguous cases, liver biopsy should be considered [2].

Due to the prevalence of obese and overweight individuals, there are some red flags to consider in order to correctly assess obesity-related steatosis. These are:a.not being in the classical age range for NAFLD, andb.being in the classical age range but with abnormalities in liver tests or hepatic ultrasounds not corresponding satisfactorily to a weight loss obtained by dietary/physical activity measures. 

As summarized in Figure 1,
a.NAFLD usually does not occur in extremely young children (those younger than 3 years) and it is rare in children younger than 10 years. In these cases, however, the coexistence of central adiposity (waist circumference >90th percentile), elevated body mass index(BMI) (>85th percentile), clinical or laboratory signs of insulin resistance, family history of NAFLD or type 2 diabetes mellitus (DM), and a sedentary lifestyle may be indicative of NAFLD [2].b.early onset liver steatosis (for those <5 years of age) or clinical phenotypes of acute liver failure, neonatal conjugated jaundice, or large organomegaly are indicative of liver diseases other than NAFLD.

## 3. Genetic and Metabolic Disorders

A large group of genetic and metabolic disorders can clinically present with fatty liver and/or hypertransaminasemia.

Although inborn errors of metabolism (IEM) can present in the neonatal period with acute symptoms, sometimes they may manifest only later in infancy or childhood with a NAFLD-like picture and a lack of other specific signs and symptoms. In these cases there is a high risk of misdiagnosis. Below we summarize some examples of this.

### 3.1. Urea Cycle Disorders

Urea cycle disorders have very variable clinical presentations which may include not only severe forms with neonatal period onset but also milder forms with onset in childhood or adulthood, depending on the different residual activity of the enzyme involved. It is typical to find microvesicular steatosis on liver histological examination [3]. In particular, given that ornithine transcarbamylase (OTC) deficiency is an X-linked partially dominant disorder, heterozygous females may have a mild form of the disease without an obvious previous history of hyperammonemia [4].

### 3.2. Citrin Deficiency

This is a rare autosomal recessive urea cycle defect. Beyond the neonatal age, when it is prevalently characterized by cholestasis, citrin deficiency may display as a diffusely fatty liver, which is histologically similar to NAFLD. A protein-rich and/or lipid-rich food preference, along with an aversion to carbohydrate-rich foods, and, later on, recurrent hyperammoniemia with neuropsychiatric symptoms (so-called citrullinemia type 2), are useful diagnostic red flags [5,6].

### 3.3. Glycogenosis (Glycogen Storage Diseases(GSD))

GSD is a form of IEM that causes abnormal synthesis of glycogen, resulting in its excess and accumulation in the organs. In the hepatic forms (GSD III, IV, VI, IX, and XI), dyslipidaemia contributes to mixed macro- and micro-vesicular steatosis. In some forms (e.g. types VI and IX), accompanying metabolic dysfunctions like hypoglycemia and acidosis are minimal and, especially in obese patients, the diagnosis may be confused with NAFLD [7,8].

### 3.4. Hereditary Fructose Intolerance

Hypertransaminasemia, hepatomegaly and ultrasonographic bright liver constitute the typical debut of hereditary fructose intolerance after the first few months of life. In this case a fructose-free diet results in an improvement of liver enzymes and of the steatotic pattern [9]. In some instances, namely those involving individuals who can defend themselves by spontaneous avoidance of noxioussugar, clinically asymptomatic fatty liver may be more easily misidentified as NAFLD.

### 3.5. Congenital Disorders of Glycosylation (CDG)

CDG are a group of disorders that result in defective synthesis of glycoproteins or glycolipids. In some forms, clinically paucisymptomatic, isolated steato-fibrotic liver involvement may be the only or prevalent indication [10,11].

### 3.6. Cholesteryl Ester Storage (CESD)

CESD and Wolman disease are rare genetic disorders characterized by a deficiency of the lysosomal acid lipase enzyme (LAL). With Wolman disease there is no residual activity of the enzyme. Clinically, it is characterized by a typical symptomatic triad of failure to thrive, hepatosplenomegaly, and steatorrhea. In CESD, however, there is some residual activity of the lysosomal acid lipase, and therefore symptoms may be less severe and more heterogeneous, causing some individuals to remain undiagnosed until adulthood. Abnormal storage of cholesteryl esters results in hepatomegaly, hepatic steatosis, and fibrosis, which can rather covertly lead to micronodular cirrhosis [12,13].

### 3.7. Abetalipoproteinemia/hypobetalipoproteinemia

Abetalipoproteinemia and hypobetalipoproteinemiaare caused by mutations of genes responsible for the transport of lipoproteins. These diseases are characterized by low levels of apolipoprotein B and low-density lipoprotein cholesterol. Clinical phenotypes range from an asymptomatic picture to failure to thrive, malabsorption, diarrhea, neurological and neuromuscular signs (spinocerebellar degeneration), and retinopathy. Many individuals with abetalipopreteinemia or hypobetalipoproteinemia develop an abnormal accumulation of fat in the liver, probably resulting from the defective synthesis and processing of apolipoprotein B. This leads to a failure in the assembly of very-low-density lipoprotein which cannot then be released from hepatocytes [14,15].

### 3.8. Alpha-1 Antitrypsin Deficiency(A1AT)

A1AT is characterized by a mutation in the *SERPINA1* gene which leads to a shortage of alpha-1 antitrypsin or to an abnormal form of the protein in the liver and/or in the lungs. In 10–15% of cases there is hepatic involvement because A1AT is not secreted properly and therefore accumulates in the liver. The clinical presentation of this liver disease is variable.In neonatal age its most frequent manifestations are hepatitis and cholestatic jaundice; in children or in young adults a chronic liver disease is more common. It is not directly associated with fatty liver, but their coexistence may aggravate their respective clinical courses [16].

### 3.9. Wilson’s Disease (WD)

Wilson’s disease has an estimated prevalence of one in 30,000, but it is higher in China, Japan, and Sardinia in Italy [17,18]. WD is characterized by copper accumulation in the body, mainly in the liver and the brain. It is caused by mutations of the *ATP7B* gene that lead to dysfunction of the copper-transporting enzyme P-type ATPase. This enzyme is responsible for transporting copper into bile and incorporating it into ceruloplasmin. The protein is synthesized by the liver and is the major copper carrier in the blood. A reduction of ceruloplasmin in the plasma is a sign of WD. Although copper starts to accumulate in the liver from the first few years of life onwards, the only signs being fatty liver and hypertransaminasemia, a symptomatic and progressive liver disease appears later. During adolescence or early adult life, untreated patients present neurological and psychiatric manifestations [17,18]. Hypoceruloplasminemia (<20 mg/dL) increased basal and post-penicillamine challenge urinary copper excretions are useful screening tests which require confirmative molecular tests or, in select cases, liver biopsies with tissue copper measurement [18]. A diagnosis of WD is usually straightforward in children with advanced liver disease, as the classical biochemical features of disturbed copper metabolism are usually present. As pointed out by the ESPGHAN, establishing a diagnosis of WD in young asymptomatic children with mild liver disease is, however, often challenging given that their ceruloplasmin levels and urinary copper excretions may be normal, and Kayser-Fleischer rings absent [19]. In clinical practice, the degree of steatosis is correlated with hepatic parenchymal copper concentration [14].

### 3.10. Cystic Fibrosis (CF)

CF is the most frequent serious autosomal recessive disorder in Caucasians presenting with a multiorgan involvement. It results from mutations within the CF transmembrane conductance regulator (CFTR). CF related liver disease (CFLD) has become one of the leading causes of morbidity and mortality in CF patients. Hepatic steatosis is sometimes associated with CF. It does not seem related to CFTR secretory defects but is likely related to selective nutritional deficiencies and an altered phospholipid metabolism [20]. In the infantile period, cholestasis is the predominant finding, although periportal macrovesicular steatosis may also be encountered [14]. That said, in rare cases CF can present only with long-term isolated or prevalent hepatic symptoms [21].

### 3.11. Shwachman-Diamond Syndrome (SDS)

Hepatomegaly and aminotransferase elevation is observed in most patients with Shwachman-Diamond syndrome. As with CF, liver damage can sometimes be the initial manifestation of this disease. The mechanism of liver damage is not completely clear [22], although autoimmune-like liver disease and antigliadin antibody positive inflammatory enteropathy might possibly be involved [23].

### 3.12. Down Syndrome

Non-alcoholic fatty liver disease is a frequent comorbidity of Down syndrome, probably due to poor physical activity [24,25]. In these patients, obesity and obstructive sleep apnea syndrome-related nocturnal hypoxia, by inducing oxidative stress in the liver, may represent additional risk factors triggering NAFLD and its progression to more severe forms [26,27].

### 3.13. Turner Syndrome

Liver involvement is also a frequent issue for patients with Turner syndrome. Hypertransaminasemia, steatosis, steatofibrosis and steatohepatitis are the most frequently reported liver-related conditions [28]. The causes of liver damage are very heterogeneous, as they range from obesity and insulin resistance, to an increased predisposition to autoimmunity, to hepatotoxicity from substitutive hormonal therapies [29].

## 4. Gastrointestinal and Nutritional Diseases

### 4.1. Celiac Disease

Celiac disease may be associated with hypertransaminasemia and fatty liver disease [30,31]. “Celiac hepatitis” is the most common hepatopathy in these patients and is characterized by minimal and non-specific histological liver lesions. These conditions are generally reversible after just a few months following the start of a gluten-free diet [32]. In cases of fatty liver which are not responsive to a gluten-free diet it is necessary to exclude other causes of liver damage, including obesity [33] and other diseases affecting the liver, such as autoimmune hepatitis, autoimmune cholangitis, and overlap syndrome [31,34].

### 4.2. Inflammatory Bowel Diseases (IBD)

IBD is associated with various hepatobiliary disorders. With IBD, the prevalence of liver dysfunction rises from 3% to 50%, according to definitions used in different studies. Fatty liver is considered the most common hepatobiliary complication in inflammatory bowel diseases while primary sclerosing cholangitis is the most specific complication [35]. However, it must be recognized that fatty infiltration can be directly related to the severity of IBD, and to malnutrition and corticosteroid use. As IBD improves and a better nutritional status is maintained, the fatty liver infiltration may improve and be reversible [36,37].

### 4.3. Malnutrition

Some systemic diseases such as severe malnutrition, cachexia and anorexia nervosa can lead to fatty liver. Starvation causes hepatocyte injury and death, leading to a rise in aminotransferases. Weight loss can produce mild elevation of transaminases, but alanine aminotransferase (ALT) and aspartate aminotransferase (AST) elevation can also occur early in the course of refeeding if dextrose calories are excessive, and is referred to as steatosis. In fact, during starvation, US typically reveals that the liver is small in size, while in “refeeding hepatitis”, US indicates an enlarged fatty liver [38,39].

## 5. Endocrine Diseases

### 5.1. Diabetes Mellitus

The liver is intensely involved in glucose metabolism and is therefore closely related to DM pathophysiology. The prevalence of NAFLD for those with Type 1 Diabetes is unknown in children. A retrospective analysis of a cohort of pediatric patients with Type 2 Diabetes revealed a 48% prevalence of elevated serum aminotransferases versus a 10% rate reported for obese teens without diabetes. Elevation of transaminases seems unrelated to age, body mass index, glycemic control, blood lipids, and diabetic therapy [40]. NAFLD in diabetic children should be distinguished from DM-associated hepatic glycogenosis (so-called Mauriac syndrome) which occurs in individuals with poorly controlled diabetes (particularly in those with Type 1) and is potentially reversible with sustained glycemic control.

### 5.2. Hypothyroidism

A reduction in thyroid function can lead to NAFLD due to reduced thyroid hormone signaling in the liver, which results in decreased hepatic lipid utilization and secondary lipid accumulation. Steatotic liver tissue leads to hepatic insulin resistance, which, along with lowered insulin secretion, leads to increased serum glucose levels and finally to de novo lipogenesis in the liver. High values of thyroid-stimulating hormone (TSH) have been considered to be involved in the pathogenetic process of NAFLD/NASH, independent of the values of thyroid hormones. Levothyroxine supplementation has clear benefits for NAFLD. A correlation between NAFLD and thyroid autoantibodies positivity in patients with normal TSH warrants consideration as well [41]. From this evidence it seems important to measure aminotransferase activity in each patient with hypothyroidism, and vice versa.

### 5.3. Hypothalamic Diseases

The hypothalamus plays a crucial role in the homeostasis of the synthesis of anterior pituitary hormones. Its damage results therefore in dysregulation of multiple pituitary hormones and subsequent complications such as hypothalamic obesity. Hypothalamic obesity with extremely rapid NAFLD onset, sometimes even before obesity worsens, is an important complication observed in patients with brain tumors before and/or subsequent to their undergoing resection or radiation therapy. Hypothalamic damage causes suppression of the sympathetic nervous system, resulting in compromised voluntary energy expenditure, and decreased insulin sensitivity due to increased vagal efferent stimulation secondary to hypothalamic injury. High serum leptin levels in hypothalamic obesity suggests leptin resistance [42,43].

## 6. Hepatitis

### 6.1. Chronic Viral Hepatitis

Micro-and macro-vesicular steatosis is a common finding in children with chronic viral hepatitis. In particular, the relationship between chronic hepatitis C (CHC) and liver steatosis has been well documented. The latter occurs in about a quarter of children with chronic hepatitis C but is less common than in adults. Studies show a major association between fatty liver and HCV genotype 3, while in patients infected with non-3 HCV genotypes the risk of steatosis is connected toa higher BMI.

Liver steatosis in hepatitis B virus (HBV)-infected children has been less analyzed than in HCV patients. In different studies the prevalence of liver steatosis in HBV-infected children ranges from 4% to 13%. Presence of steatosis in these patients seems to be associated more with metabolic factors (obesity, metabolic syndrome, hyperglycemia and hyperglycemia, and increased blood pressure) than with viral determinants [44].

HIV-infected persons are about twice as likely to develop steatosis, and superimposed HEV infection has been shown to be related to more severe liver abnormalities of unclear etiology in this particular population [45].

### 6.2. Autoimmune Hepatitis (AIH)

AIH in conjunction with steatosis or steatohepatitis is not a rare condition, affecting10–30% of patients with AIH (the same rate as the general population). The relationship between these diseases is not well known, but in patients with NAFLD/NASH a high prevalence of antinuclear and/or anti-smooth muscle antibodies has been demonstrated. Evidence suggests that AIH coincident with NAFLD/NASH may lead more rapidly to liver cirrhosis, and patients are more likely to develop poor responses to corticosteroids [46,47].

## 7. Iatrogenic Causes

Exposure to some toxic substances and drugs has been associated with the development of liver disease with fatty liver and/or hypertransaminasemia, thus requiring an exclusion diagnosis using an accurate medical history when making a diagnosis of NAFLD.

### 7.1. Alcohol Consumption

Increase in alcohol consumption by adolescents is likely becoming a major cause, after obesity, of liver damage in young adults. Binge drinking is the most common pattern of alcohol consumption among high-school youth and is more frequent in males than in females [48]. Elevated serum gamma-glutamyl transferase (GGT) levels, an AST/ALT ratio >1, and/or an increase in mean corpuscular volume may help in identifying excessive drinking [49].

### 7.2. Toxic Substances

Long-term exposure to toxic substances such as ecstasy, cocaine, solvents, and pesticides may be relevant to the differential diagnosis process.

### 7.3. Drug Toxicity

While in adults there are several well-established tools which may be used to quantitatively assess causality in cases with suspected drug-induced liver injury (DILI) and herb-induced liver injury (HILI) (e.g. the Roussel Uclaf Causality Assessment Method (RUCAM)) [50], these tools have not been fully validated in pediatrics. The Drug-Induced Liver Injury Network (DILIN) Prospective Study, a longitudinal multicenter study designed to determine the etiologies, risk factors, and outcomes of suspected DILI, may therefore be useful to pediatricians for navigating the large amount of possible causes for this age group [51]. It is important to consider that the relationship between DILI and NAFLD may be reciprocal; drugs can cause NAFLD by acting as steatogenic factors, and pre-existing NAFLD could be a predisposing condition for certain drugs to cause DILI [52,53]. Regarding steatosis, as shown in Table 1, long-term therapy with glucocorticoids, methotrexate, tetracycline, amiodarone, nucleoside analogues, aspirin, and antiretroviral drugs is to be considered. Corticosteroids are probably those most commonly associated with macrovesicular steatosis. They act by stimulating the transcription of lipogenic enzymes and inhibiting fatty acid beta oxidation enzymes. Chronic therapy with valproic acid is associated with an increased development of microvesicular steatosis, especially in patients who are initially overweight and have features of metabolic syndrome [14,18,54].

## 8. Myopathies

Genetic muscular disorders such as dystrophinopathies with muscular origin hypertransaminasemia may occasionally hide a sedentary based obesity-related fatty liver. Unfortunately, sometimes children may undergo liver biopsies without a prior exclusion of the muscular causes of fatty liver revealed by high blood levels of muscular enzymes. In these cases a misdiagnosis of NAFLD is possible [55,56].

In summary, the causes hitherto mentioned may schematically be classified as shown in Table 1.

## 9. Discussion and Conclusions

In the diagnostic work up of NAFLD, anthropometric estimations and clinical features (BMI, waist circumference, and signs of insulin resistance (e.g. acanthosis nigricans) should be considered when predicting the risk of NAFLD. Signs such as belonging to a certain ethnic group (e.g. Hispanic) [57] or a dietary history revealing large amounts of fructose intake may aid in strengthening a presumptive diagnosis of obesity- or diet-related NAFLD [58]. First line approaches should include liver ultrasonography and simple laboratory tests [2].

Ultrasonographic exams (compared to liver biopsies) have a sensitivity ranging from 60% to 96% and a specificity ranging from 84% to 100%, depending on the grade of steatosis severity [59,60]. When considering other available imaging modalities which can obtain more accurate quantitative evaluations of fat content, a recent meta-analysis has confirmed the superiority of MRI and MRI spectroscopy (MRS) [61,62]. MR elastography has also been used to accurately assess hepatic fibrosis [63]. However, equipment for these techniques are not present in every hospital and are available mainly in specialized centers.

Standard liver function tests represent an option worth suggesting among laboratory exams [64], although it should be noted that cases of normal transaminases-NAFLD are not infrequent and may therefore be misleading [2,65].

When the patient is not responsive to physical-dietary interventions or when weight loss is not followed by transaminases/ultrasonographic improvement it is prudent to exclude the most common etiologies of fatty liver and hypertransaminasemia other than obesity-related NAFLD by measuring a number of indices with a stepwise approach as summarized in Table 2.

More invasive investigations (the third step), including liver biopsies, could be justified, especially when these exams are inconclusive. Liver biopsies are in fact the gold standard for diagnosis in patients with fatty liver because in addition to its being able to distinguish NAFLD from NASH, it can also exclude or address the identification of other steatotic liver diseases (e.g., in general, IEM has a microvacuolar pattern). Yet, indications that liver biopsies are an essential test are still under discussion and there is no consensus to formulate precise guidelines [2]. In fact, it is an expensive and invasive technique associated with possible complications, and, due to an unpredictable sampling variability, it is not exempt from false negative results [66,67]. For these reasons, noninvasive biological markers, scoring systems, and imaging modalities (so called “liquid biopsies”) are being developed and investigated more and more to better assess NAFLD patients [68].

As previously mentioned, a high prevalence of obesity may increase the risk of NAFLD overdiagnosis and possibly deprive the patient of a specific curative diet or drug therapy for another treatable condition. For example, a gluten free diet for patients with celiac disease as well as a fructose free diet for those with hereditary fructose intolerance will allow patients to be cured, have no symptoms, and have a normal life expectancy. Several other conditions also have effective therapies which can improve the liver health of young patients (e.g. copper chelation for Wilson’s disease and steroids for autoimmune hepatitis).

In conclusion, independently of patient BMI, pediatric fatty liver requires keen evaluation by considering several conditions for those of all ages before relying on a diagnosis of NAFLD.

## Figures and Tables

**Figure 1 children-05-00169-f001:**
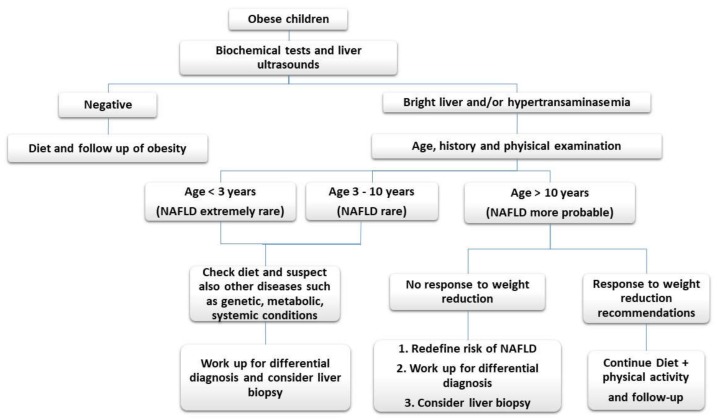
Algorithm for differential diagnosis of fatty liver disease in children.

**Table 1 children-05-00169-t001:** Fatty liver etiologies to be considered against a diagnosis of pediatric non-alcoholic fatty liver disease (NAFLD)/non-alcoholic steatohepatitis (NASH).

Gastrointestinal/Nutritional/ Endocrine/Hepatic Causes	Genetic and Metabolic Causes	Toxics and Drugs
Celiac disease	α- and β-oxidation defects	Cocaine
Inflammatory bowel disease	Abeta or hypobetalipoproteinemia	Ecstasy
Anorexia nervosa	Cholesterol ester storage disease	Ethanol
Obesity	Citrin deficiency	Pesticides
Severe malnutrition	Cystic fibrosis	Solvents
Diabetes mellitus type 1	Glycogen storage disease	Amiodarone
Hypothalamic-pituitary disorders	Hereditary fructose intolerance	Antiretroviral drugs
Hypothyroidism	Mitochondrial and peroxisomal defects	Aspirin
Polycystic ovary syndrome	Shwachman-Diamond syndrome	Glucocorticoids
Autoimmune hepatitis	Turner syndrome	Methotrexate
Viral hepatitis	Urea cycle disorders	Sodium valproate
	Wilson’s disease	Tetracycline

**Table 2 children-05-00169-t002:** Laboratory workup in children with fatty liver.

1st Step	2nd Step
Blood counts and standard liver function tests (AST, ALT, GGT, coagulation, bilirubin, total protein and electrophoresis, total Ig)	Clinically oriented hormonal tests (e.g. cortisol)
Lipid profile, glucose, insulin (eventually OGTT)	Serum copper, 24h urinary copper
FT3, FT4,TSH	Sweat test/molecular test for CFTR
EMA, tTgasiIgA	AMA, SMA, LKM, LC1
Viral markers (HBV, HCV)	A1-antitrypsin
Ceruloplasmin	Amino and organic acids, acyl carnitine profile, serum lactate, ammonium, CDG and LAL test, urinary reducing substances

The abbreviations used above areALT: Alanine aminotransferase;AMA: Anti-mitochondrial antibodies;AST: Aspartate aminotransferase;CDG: Congenital disorders of glycosylation; CFTR: Cystic fibrosis transmembrane conductance regulator; EMA: Anti-endomysial antibodies; FT3: Free T3; FT4: Free T4; GGT: Gamma-glutamyl transferase; HBV: Hepatitis B virus; HCV: Hepatitis C virus; Ig: Immunoglobulin; LAL: Lyposomal acid lipase; LC1: Anti-liver cytosol antibodies;LKM: Anti-liver-kidney microsomal antibodies; OGTT: oral glucose tolerance test; SMA: Anti-smooth muscle antibodies; TSH: Thyroid-stimulating hormone; tTgasi: tissue Transglutaminase.

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
