# Peer review of "Pediatric Fatty Liver and Obesity: Not Always Just a Matter of Non-Alcoholic Fatty Liver Disease"

_children, 2018, doi:10.3390/children5120169_

Round 1

Reviewer 1 Report

This is a very well written and thorough review of NAFLD and other diagnostic possibilities to be considered when making the diagnosis.

I have no edits to the text submitted but have a few suggested additions

1.       I would add a paragraph or two on the pathogenesis of NAFLD, particularly the potential role of fructose in the development of the disease and restriction may have a role in treatment.   References to consider adding:

a.      Vos MB, Lavine JE. Dietary fructose in nonalcoholic fatty liver disease. Hepatology. 2013 Jun 1;57(6):2525-31.

b.     Jin R, Le NA, Liu S, Farkas Epperson M, Ziegler TR, Welsh JA, Jones DP, McClain CJ, Vos MB. Children with NAFLD are more sensitive to the adverse metabolic effects of fructose beverages than children without NAFLD. The Journal of Clinical Endocrinology & Metabolism. 2012 Jul 1;97(7):E1088-98.

c.      Lim JS, Mietus-Snyder M, Valente A, Schwarz JM, Lustig RH. The role of fructose in the pathogenesis of NAFLD and the metabolic syndrome. Nature reviews Gastroenterology & hepatology. 2010 May;7(5):251.

2.      I would add a section on the epidemiology and how certain ethnic groups such as children who are Hispanic have a high prevalence

3.      For diagnosis I might consider adding MRI elastography as a potential tool

a.      Xanthakos SA, Podberesky DJ, Serai SD, Miles L, King EC, Balistreri WF, Kohli R. Use of magnetic resonance elastography to assess hepatic fibrosis in children with chronic liver disease. The Journal of pediatrics. 2014 Jan 1;164(1):186-8.

Author Response

Pediatric Fatty Liver and Obesity: Not Always Only A Matter of Non-Alcoholic Fatty Liver Disease

We thank the Reviewer for their insightful and pertinent comments .

REVIEWER 1

This is a very well written and thorough review of NAFLD and other diagnostic possibilities to be considered when making the diagnosis.

I have no edits to the text submitted but have a few suggested additions

1. I would add a paragraph or two on the pathogenesis of NAFLD, particularly the potential role of fructose in the development of the disease and restriction may have a role in treatment.   References to consider adding:

a. Vos MB, Lavine JE. Dietary fructose in nonalcoholic fatty liver disease. Hepatology. 2013 Jun 1;57(6):2525-31.

b. Jin R, Le NA, Liu S, Farkas Epperson M, Ziegler TR, Welsh JA, Jones DP, McClain CJ, Vos MB. Children with NAFLD are more sensitive to the adverse metabolic effects of fructose beverages than children without NAFLD. The Journal of Clinical Endocrinology & Metabolism. 2012 Jul 1;97(7):E1088-98.

c. Lim JS, Mietus-Snyder M, Valente A, Schwarz JM, Lustig RH. The role of fructose in the pathogenesis of NAFLD and the metabolic syndrome. Nature reviews Gastroenterology & hepatology. 2010 May;7(5):251.

A1. We added a paragraph related to the importance of inquiring on the fructose dietary intake as a red flag for probable diet-related NAFLD, with suggested reference (a) (page 8, Discussion section, line 307-308; refs 57)

===

2. I would add a section on the epidemiology and how certain ethnic groups such as children who are Hispanic have a high prevalence

A2. We added a paragraph related to Epidemiology (section 2 NAFLD, page 2, line 50-51) and referred to the importance of considering the ethnicity as a red flag for the possible presence of NAFLD , with a corresponding  reference (Discussion section , page 8, line 306; ref 58)

====

3. For diagnosis I might consider adding MRI elastography as a potential tool

a. Xanthakos SA, Podberesky DJ, Serai SD, Miles L, King EC, Balistreri WF, Kohli R. Use of magnetic resonance elastography to assess hepatic fibrosis in children with chronic liver disease. The Journal of pediatrics. 2014 Jan 1;164(1):186-8 ____

A3. We have added a more recently referenced paragraph on the newest MRI imaging and liquid biopsy tools  (Discussion Section, page 9, line 314-315; refs 64, and 63)

Reviewer 2 Report

This is a careful review article on an important clinical issue in children. Some improvements are suggested.

Minor points:

1. Tables 1 and 2 are not suitable for clinical practice. Instead, I am missing an informative table with clear parameters how other diagnoses can be ruled out. A similar problem applies for example to drug induced liver injury, a respective list is published: Danan G, Teschke R. RUCAM in drug and herb induced liver injury: An update. Int J Mol Sci 2016, 17,14. You may use part of this list and adapt to children.Take care of Wilson disease, its diagnostic workup is often insufficient. What about HEV?

2. A more critical discussion is needed for liver biopsy as this is an invasive procedure and only justified if no other approaches are available.

3. Line 248: Myopathies..

4. How do you diagnose DILi superimposed on NAFLD and NASH? Do you suggest the updated RUCAM? Please discuss this point in detail and reference current papers.

Author Response

Pediatric Fatty Liver and Obesity: Not Always Only A Matter of Non-Alcoholic Fatty Liver Disease

We thank the Reviewer for their insightful and pertinent comments .

REVIEWER 2

This is a careful review article on an important clinical issue in children. Some improvements are suggested.

Minor points:

1. Tables 1 and 2 are not suitable for clinical practice. Instead, I am missing an informative table with clear parameters how other diagnoses can be ruled out.

A similar problem applies for example to drug induced liver injury, a respective list is published: Danan G, Teschke R. RUCAM in drug and herb induced liver injury: An update. Int J Mol Sci 2016, 17,14. You may use part of this list and adapt to children.Take care of Wilson disease, its diagnostic workup is often insufficient.

What about HEV?

A1.1) We have added a paragraph on DILI (Section 7.3, page 7, line 277-285, refs 50-51,52,53) and have added information on the WD diagnosis difficulties (section 3.9, WD, Page 4, line 150-155; ref 19)

A1.2) We have added a paragraph on HEV (Section 6.1 Viral hepatitis, page 7, 253-255, ref. 45)

===

2. A more critical discussion is needed for liver biopsy as this is an invasive procedure and only justified if no other approaches are available.

A2) We have edited the corresponding paragraph (page 9, Discussion, line 333-342)

===

3. Line 248: Myopathies..

A3) We have corrected the typo (section 8; page 8, now line 293)

===

4. How do you diagnose DILi superimposed on NAFLD and NASH? Do you suggest the updated RUCAM? Please discuss this point in detail and reference current papers.

A4) see A1.1